# Blueberry Prevents the Bladder Dysfunction in Bladder Outlet Obstruction Rats by Attenuating Oxidative Stress and Suppressing Bladder Remodeling

**DOI:** 10.3390/nu12051285

**Published:** 2020-05-01

**Authors:** Nozomu Miyazaki, Ryota Katsura, Koki Hamada, Tatsuo Suzutani

**Affiliations:** Department of Microbiology, School of Medicine, Fukushima Medical University, Fukushima 960-1295, Japan; m161031@fmu.ac.jp (R.K.); m151099@fmu.ac.jp (K.H.); suzutani@fmu.ac.jp (T.S.)

**Keywords:** blueberry, Overactive Bladder Symptoms, Bladder Outlet Obstruction, rat model, functional food

## Abstract

Various berries demonstrate antioxidant activity, and this effect is expected to prevent chronic diseases. We examined whether a diet containing blueberry powder could prevent the development of bladder dysfunction secondary to bladder outlet obstruction (BOO). Eighteen 8-week-old male Sprague-Dawley rats were randomly divided into three groups: Sham (sham operated + normal diet), N-BOO (BOO operated + normal diet) and B-BOO (BOO operated + blueberry diet). Four weeks after BOO surgery, the N-BOO group developed bladder dysfunction with detrusor overactivity. The B-BOO group showed significantly improved micturition volume and micturition interval. The urinary levels of 8-hydroxy-2′-deoxyguanosine (8-OHdG) and malondialdehyde (MDA) were measured as oxidative stress markers. In the N-BOO group, 8-OHdG increased 1.6-fold and MDA increased 1.3-fold at 4 weeks after surgery, whereas the increase in 8-OHdG was significantly reduced by 1.1-fold, despite a similar increase in MDA, in the B-BOO group. Bladder remodeling was confirmed due to bladder hypertrophy, fibrosis and increased connexin43 expression in the N-BOO group, but these histological changes were reduced in the B-BOO group. The intake of blueberries prevented the development of bladder dysfunction secondary to BOO. This effect seems to be related to antioxidation and the inhibition of bladder remodeling.

## 1. Introduction

Blueberries are widely consumed as a health food around the world. The bioactive components of blueberries demonstrate anti-oxidative, anti-inflammatory and anti-cancer effects [1,2,3,4]. Previous studies suggest that the major bioactive components of blueberries are anthocyanins, phenolic acids, ascorbic acid, flavonol and tannins [1]. Anthocyanins extracted from blueberries can prevent carcinogenesis and reduce the risk of cancer recurrence [2,5,6], while blueberries rich in anthocyanins increase anti-inflammatory cytokine levels and attenuate oxidative stress [3,7,8,9,10]. Blueberry juice also ameliorates hepatic fibrosis in rats [11]. The functional components of blueberries are expected to be useful in both preventing and treating chronic diseases caused by oxidative stress and chronic inflammation. In the future, blueberries may be used as prophylactic functional foods for chronic diseases such as cancer, diabetes and degenerative diseases, in addition to their use in the development of therapeutic agents based on these components [4,5,6].

Lower urinary tract dysfunction (LUTD) refers to impairments in urine storage and voiding, including bladder dysfunction such as detrusor overactivity (DO) and underactivity, and induces lower urinary tract symptoms (LUTS) such as storage, voiding and post-micturition symptoms. More than half of the middle- and old-aged populations have LUTS [12,13,14], and it is a concern that such symptoms reduce the quality of life [15,16,17].

Overactive bladder (OAB) is a syndrome with various etiologies characterized by urgency, frequent urination and nocturia [18]. OAB can be classified into neurogenic OABs and non-neurogenic OABs. In this study, we focused on one of the non-neurogenic OABs induced by BOO, with benign prostatic enlargement (BPE). The mechanisms underlying OAB with associated BPE are considered to involve mechanical obstruction of the urethra and functional obstruction due to smooth muscle contraction from the bladder neck to the prostatic urethra resulting from an enlarged prostate. These changes lead to extension, increased pressure and bladder ischemia, and also cause irritability of the bladder smooth muscle and hyperactivity of the afferent nerve with denervation supersensitivity [19]. Oxidative stress accumulated by ischemia is also considered to be one of the underlying mechanisms [20]. Moreover, bladder remodeling due to smooth muscle layer fibrosis and increased expression of connexin (Cx) 43 interferes with bladder smooth muscle coordination, resulting in abnormal bladder contraction, and it is considered to be one of the underlying causes of bladder smooth muscle irritability [21,22].

We hypothesized that components within blueberries could prevent the development of bladder dysfunction by attenuating chronic oxidative stress. Therefore, in this study, we investigated whether the intake of a diet containing dried blueberry powder in rats with bladder outlet obstruction could prevent or attenuate the development of bladder dysfunction.

## 2. Materials and Methods

### 2.1. Animals and Experimental Design

The current study used BOO rats, a model of lower urinary tract obstruction. The experimental protocol was reviewed and approved by the Fukushima Medical University Animal Ethics Committee (permission No.28093).

Eighteen 8-week-old male Sprague Dawley rats weighing 260 to 300 g were used. The rats received a normal diet or a normal diet containing 2% freeze-dried Canadian wild blueberry powder (Blueberry diet) for 7 days preoperatively. The rats then underwent BOO or sham surgery at 9 weeks old. From the day of the operation, the BOO rats continued to receive a normal diet (N-BOO group) or a Blueberry diet (B-BOO group) for 4 weeks after surgery. Sham-operated rats also continued to receive a normal diet (Sham group) for 4 weeks.

During week 4 postoperatively, urine was collected from the rats housed in a metabolic cage for a 24-hour period to determine the urinary concentration of creatinine and oxidative stress markers, as described below. Continuous cystometry was carried out on conscious rats from each group. After cystometrogram recording, rats were sacrificed and the bladder was removed and weighed. The bladders were then examined by histological and Western blot analyses.

### 2.2. BOO Surgery

Nine-week-old rats were anesthetized with 2% isoflurane. BOO was created by incomplete ligation of the urethra using a standardized method [23]. Sham-operated rats underwent similar surgery but no obstruction was created.

### 2.3. Continuous Cystometry

On day 3 prior to continuous cystometry, each rat was anesthetized with 2% isoflurane. INTRAMEDIC™ PE-50 polyethylene tubing (BECTON DICKINSON Japan, Tokyo, Japan) was inserted into the bladder dome and the other end of the tube was brought out through the subcutaneous tunnel in the back. Three days later, continuous cystometry was performed in conscious rats without restraint in a metabolic cage [24]. The bladder tube was connected through a T-tube to a pressure transducer for recording bladder pressure and to an infusion pump for continuous saline infusion into the bladder. An analytical balance was placed under the metabolic cage. The transducer and balance were connected to a PowerLab^®^ system to record bladder pressure and micturition volume as saline was infused into the bladder at a rate of 10 mL per hour.

### 2.4. Measurement of Urinary Oxidative Stress Markers

We determined the levels of urinary 8-hydroxy-2′-deoxyguanosine (8-OHdG) and malondialdehyde (MDA) as markers of DNA damage and lipid peroxidation, respectively. To evaluate the urinary levels of these markers, the urine collected from rats was centrifuged at 5,000 x g for 10 min. The 8-OHdG, MDA and creatinine in the resulting supernatant were determined using a New 8-OHdG ELISA Kit (Japan Institute for the Control of Aging, Shizuoka, Japan), a NWLSS^TM^ malondialdehyde Assay (Northwest Life Science Specialties, Vancouver, British Columbia, Canada) and LabAssay^TM^ Creatinine Kit (Wako, Osaka, Japan), respectively.

### 2.5. Histological Analysis

Bladders from 6 rats per group were immersed in 10% neutral buffered formalin, embedded in paraffin and cut into 5 μm sections. Bladder sections were stained with hematoxylin–eosin, and Masson trichrome (Muto Pure Chemicals, Tokyo, Japan). The percentage of collagen in the bladder muscle layer was calculated for 4 high power fields as the sum of the blue-stained area divided by the sum of all red- and blue-stained areas.

### 2.6. Western Blot Analysis

We determined the expression levels of Cx43 and connective tissue growth factor (CTGF) protein in the bladder. Bladder extracts were prepared from a whole bladder homogenized in CelLytic^TM^ MT Cell Lysis Reagent (Sigma-Aldrich, St. Louis, MO, USA). Total protein concentration was quantified using a BCA Protein Assay Kit (Takara-Bio, Shiga, Japan) and a microplate reader. Thirty μg of homogenate protein was loaded and separated by 10% SDS-PAGE gel and transferred onto Immobilon-P transfer membranes (Merck Millipore Japan, Tokyo, Japan). The membranes were blocked for 2 h with 3% skim milk in Tris-buffered saline-Tween20 (TBS-T) at 4 ℃, and sequentially incubated with rabbit anti-Cx43 antibody (1:500, Novus Biologicals), rabbit anti-CTGF antibody (1:1000, GeneTex) or mouse anti-beta-actin antibody (1:20000, Proteintech) at 4 ℃ overnight. After washing with TBS-T, membranes were incubated for 1 hour with HRP-conjugated anti-rabbit-IgG (1:2000) or anti-mouse-IgG (1:4000) at room temperature. After washing membranes with TBS-T, protein bands were visualized with ImmunoStar^®^ Zeta (Wako). The expression levels of Cx43 and CTGF proteins per beta-actin protein of the Sham group were regarded as 1.0.

### 2.7. Data Analysis

Statistical analyses were performed using GraphPad Prism software version 6.07 (GraphPad Software Inc., San Diego, CA, USA). Results are expressed as the mean ± SE. Data were analyzed using one-way ANOVA followed by Tukey’s multiple comparison test with p <0.05 considered statistically significant.

## 3. Results

In the N-BOO group, the bladder weight increased significantly compared with that in the Sham group. However, in the B-BOO group, the increase in bladder weight was significantly lower compared to that in the N-BOO group, although it was heavier than that in the Sham group (Table 1). There were no significant differences in body weight among the three groups.

### 3.1. Cystometry

In the N-BOO and B-BOO groups, the micturition intervals were significantly shorter and micturition volumes were significantly smaller than those in the Sham group (Table 1 and Figure 1). However, the micturition interval and micturition volume for the B-BOO group were significantly greater than those in the N-BOO group. Micturition pressure in rats with BOO was significantly higher than that in the sham-operated rats, although there was no significant difference between the N-BOO and B-BOO groups. Non-voiding contractions were more frequently observed in rats with BOO than in sham-operated rats. In the N-BOO group, non-voiding contractions were more unstable than in the B-BOO group. These observations indicate that the intake of blueberry powder markedly suppressed the dysuria associated with BOO.

### 3.2. Urinary 8-OHdG and MDA

In order to clarify the mechanisms underlying the suppression of dysuria on the intake of blueberries, urinary levels of the oxidative markers, 8-OHdG and MDA, were evaluated. In the N-BOO group, the urinary 8-OHdG and MDA levels were significantly increased compared with those in the Sham group (Figure 2). However, in the B-BOO group, the 8-OHdG level was markedly decreased compared with that in the N-BOO group, although there was no significant difference in the MDA level.

### 3.3. Histological Analysis

The effect of the blueberry diet on histological changes induced by BOO was observed. In the N-BOO group, the percentage of collagen in the bladder muscle layer was markedly higher than that in the Sham group (Figure 3). In the B-BOO group, the increase in the percentage of collagen after BOO was significantly lower than that in the N-BOO group.

### 3.4. Western Blot Analysis

Histological examination indicated the expression of collagen was induced by BOO, and the increase was suppressed by the intake of the blueberry diet. Therefore, the effect of blueberries on Cx43 and CTGF were evaluated by Western blot analyses. The protein expression levels of Cx43 in the bladder in the N-BOO and B-BOO groups were significantly increased by 2.3- and 1.3-fold, respectively, compared to that in the Sham group (Figure 4). There was also a significant difference between the BOO groups. On the other hand, there were no significant differences in the CTGF expression level among the three groups.

## 4. Discussion

Blueberries have various health benefits and are consumed as a health food around the world. The bioactive components of blueberries include anthocyanins, polyphenols and vitamins, which have anti-oxidative, anti-inflammatory, anti-cancer and anti-diabetic effects [1]. We examined whether the ingestion of diets containing blueberries could replicate the health benefits in rats with BOO.

BOO rats are a pathological model of bladder dysfunction associated with BPE. The mechanism for the bladder dysfunction in this model is related to inflammation, oxidative stress, ischemia, denervation and fibrosis [20,25,26,27]. Thus, we investigated whether blueberry intake could prevent the development of bladder dysfunction in BOO rats. To our knowledge, this is the first study to report beneficial effects of blueberries for this condition.

BOO rats develop bladder dysfunction at 4 weeks after surgery, which is characterized by DO, such as increased urination frequency, shorter micturition intervals and decreased voided volume. They also show significant increases in urinary oxidative stress markers, denervation-supersensitivity and bladder smooth muscle fibrosis and hypertrophy. In the current study, the N-BOO group showed marked DO, whereas the development of DO was alleviated in the B-BOO group. In the B-BOO group, the urinary 8-OHdG level was markedly decreased compared with that in the N-BOO group. The intake of blueberries significantly reduced the increase in bladder weight, fibrosis of the smooth muscle layer and Cx43 expression level in BOO rats. Our results suggested that prophylactic blueberry intake alleviates the development of bladder dysfunction associated with lower urinary tract obstruction in rats.

Oxidative stress was assessed by the measurement of 8-OHdG, a marker of DNA oxidative damage, and MDA, a marker of lipid peroxidation. The urinary 8-OHdG and MDA levels in the N-BOO group were significantly increased. On the other hand, in the B-BOO group, the urinary 8-OHdG level was significantly decreased, while there was no change in the urinary MDA level. In a previous report, molecular hydrogen prevented the development of bladder dysfunction in BOO rats by significantly alleviating both 8-OHdG and MDA expression [28]. However, the current study suggests that only DNA oxidation is involved in the development of bladder dysfunction. Intracellular oxidative damage may play a major role in the development of bladder dysfunction in BOO rats. In addition, the anti-oxidative mechanism induced by the bioactive components of blueberries may be mainly due to scavenger action on intracellular oxidants, such as reactive oxygen species generated in the mitochondria. The reasons for the absence of any change in the urinary MDA level may be explained by the fact that only high diffusible substances, such as molecular hydrogen or fat-soluble substances such as vitamin E, can protect against oxidative damage in cell membranes. The anti-oxidative mechanisms of the bioactive components of blueberries should, therefore, be considered in future study.

In this study, the ingestion of blackcurrants, which have a strong anti-oxidative effect, was also examined. However, the development of bladder dysfunction was not alleviated by the ingestion of blackcurrants in BOO rats (data not shown). Therefore, we consider blueberries to have some specific effects in addition to the observed anti-oxidative effect. In previous reports, Wiseman et al. reported that flavonoid-rich blueberries could inhibit angiotensin-converting enzyme activity [29]. Also, Aikawa et al. reported that angiotensin receptor blockers prevented the development of DO in BOO rats [30]. These results indicate that the administration of angiotensin receptor blockers significantly suppressed bladder weight increase and bladder smooth muscle layer fibrosis in BOO rats. In other words, angiotensin receptor stimulation may be involved in the development of DO with associated bladder weight increases and bladder smooth muscle layer fibrosis in BOO rats. Our results suggest that the ingestion of blueberries could suppress angiotensin receptor stimulation and prevent DO in BOO rats. Rosin et al. also reported that CTGF is involved in fibrosis induced by angiotensin Ⅱ stimulation [31]. CTGF mediates transforming growth factor-β stimulation and promotes proliferation and extracellular matrix protein production in fibroblasts in fibrotic diseases [32]. Thus, we hypothesized that CTGF, which mediates angiotensin Ⅱ stimulation, may be involved in the fibrosis of the bladder smooth muscle layer in BOO rats. In addition, it has been reported that the expression of Cx43 is increased in association with DO in BOO rats [22,33]. Cx43, a gap junction component protein, enhances smooth muscle contraction by promoting intercellular communication and coordination [34,35], and increases the release of neurotransmitters such as adenosine triphosphate, prostaglandin Es and acetylcholine by extension stimulation and oxidative stress [36]. Qiu et al. reported that CTGF is involved in regulating Cx43 expression in cardiomyocytes [37]. Therefore, we anticipated that fibrosis of the bladder smooth muscle and increase in Cx43 expression level, induced by CTGF-mediated angiotensin Ⅱ stimulation, were related to mechanisms underlying the development of DO in BOO rats. Our results showed that there was no significant difference in CTGF expression level. However, blueberry intake significantly reduced the increase in Cx43 expression level. In addition, the B-BOO group showed a significantly reduced increase in bladder weight and fibrosis of the smooth muscle layer compared to the N-BOO group. These results suggested that blueberry intake suppressed signal transductions related to the promotion of proliferation and extracellular matrix protein production from CTGF to fibroblasts. As this effect could not be confirmed in other berries, it seems to be a specific effect of blueberries. Blueberries may be functional foods that inhibit the progress of bladder remodeling and prevent the development of bladder dysfunction in obstructed bladder. Future study on the identification of the components involved in this pathway needs to be considered.

In clinical practice, urinary urgency is important as a main complaint from patients, and it has a great influence on the QOL of patients. As this study was based on an animal model, one limitation of this study is that urinary urgency could not be evaluated. However, we believe that animal model studies can be used to investigate pathological mechanisms in detail and contribute to the discovery of new preventive and therapeutic targets.

The current study suggests that blueberries prevent the development of bladder dysfunction secondary to BOO by at least two effects; antioxidation and inhibition of bladder remodeling. The ingestion of blueberries had no potent effect via each of these effects separately; however, the two effects might exert a preventive effect by acting synergistically. Thus, daily consumption of blueberries may prevent the development of OAB in addition to chronic diseases such as cancer, diabetes and degenerative diseases.

## Figures and Tables

**Figure 1 nutrients-12-01285-f001:**
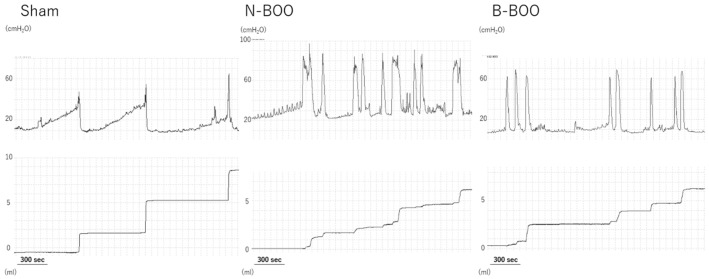
Representative cystometrogram recordings for the Sham, N-BOO and B-BOO groups.

**Figure 2 nutrients-12-01285-f002:**
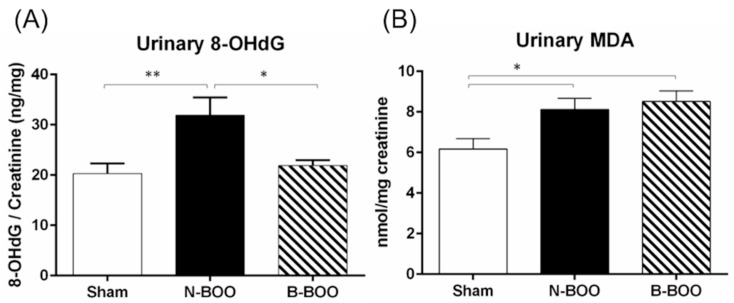
Anti-oxidative effect of blueberries on urinary 8-OHdG (**A**) and malondialdehyde (MDA) (**B**). A single asterisk indicates p <0.05. Double asterisks indicate p <0.01.

**Figure 3 nutrients-12-01285-f003:**
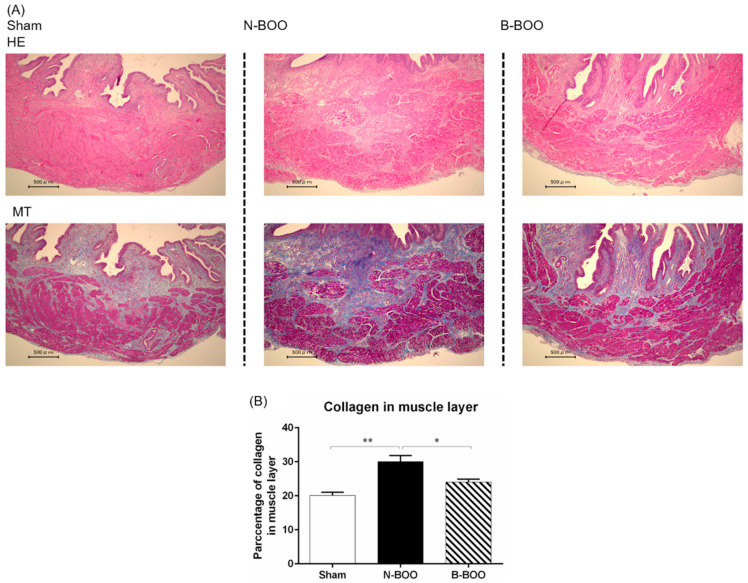
Histological analysis of the bladder tissue from Sham, N-BOO and B-BOO group rats. (**A**) Hematoxylin–Eosin staining (HE) is shown in the upper row and Masson trichrome stain (MT) in the lower row. (**B**) Mean ± SE of 6 determinations each of the percentage of collagen in muscle layer of Sham, N-BOO and B-BOO group rat bladders. A single asterisk indicates p <0.05. Double asterisks indicate p <0.01.

**Figure 4 nutrients-12-01285-f004:**
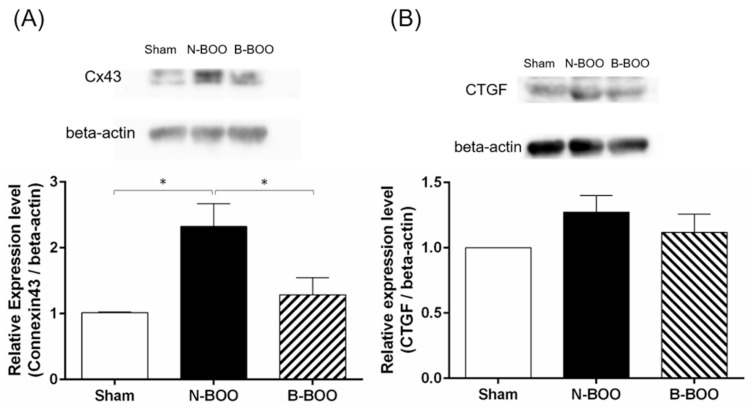
Representative Western blots of (**A**) Cx43 expression, and (**B**) connective tissue growth factor (CTGF) expression in the bladder (upper). Relative expression levels of (**A**) Cx43/beta-actin; (**B**) CTGF/beta-actin in the bladder (lower). A single asterisk indicates p <0.05.

**Table 1 nutrients-12-01285-t001:** Results in 6 sham-operated rats, 6 BOO rats received the normal diet (N-BOO) and 6 BOO rats received the Blueberry diet (B-BOO).

	Sham	N-BOO	B-BOO
Weight:			
Body (g)	496 ± 27.8	442 ± 12.4	487 ± 10.1
Bladder (mg)	238 ± 13	744 ± 155 **	405 ± 60 ^†^
Micturition:			
Pressure (cm H_2_O)	29.84 ± 4.38	56.95 ± 4.00 **	52.01 ± 2.93 **
Interval (secs)	686.49 ± 62.29	185.72 ± 25.61 **	396.77 ± 92.66 **^, †^
Volume (ml)	1.94 ± 0.21	0.38 ± 0.06 **	1.13 ± 0.15 *^, †^

A single asterisk indicates *p* <0.05. Double asterisks indicate p <0.01 vs. sham-operated rats. A single dagger indicates *p* <0.05 vs. N-BOO rats

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
