# Peer review of "Blueberry Prevents the Bladder Dysfunction in Bladder Outlet Obstruction Rats by Attenuating Oxidative Stress and Suppressing Bladder Remodeling"

_nutrients, 2020, doi:10.3390/nu12051285_

Round 1

Reviewer 1 Report

Studies on the anti-cancer properties of blueberries have been published for some time. They are very interesting. The authors write that this is the first report for bladder cancer. This is not true because a group from Taiwan led by Yang-jin Wang achieved similar conclusions. Wang's research has been published in 2008 and 2010 in Toxicological Sciences, a monthly medical research journal published by Oxford University in the United Kingdom. It would be worth referring to these results.

Author Response

Response to Reviewer 1:

We wish to express our appreciation to the Reviewer 1 for the useful comment.

Comment: Studies on the anti-cancer properties of blueberries have been published for some time. They are very interesting. The authors write that this is the first report for bladder cancer. This is not true because a group from Taiwan led by Yang-jin Wang achieved similar conclusions. Wang's research has been published in 2008 and 2010 in Toxicological Sciences, a monthly medical research journal published by Oxford University in the United Kingdom. It would be worth referring to these results.

Response: We appreciate the Reviewer’s comment. However, we believe the reviewer is mistaken on this point. Wang et al. (2008 and 2010) used bladder epithelial cancer cells to investigate changes in several signal transduction pathways after nicotine exposure. We think these reports are excellent reports that suggest the risk of smoking in the treatment of bladder cancer. However, in our study, we have focused on bladder dysfunction, such as frequent urination, associated with Benign Prostatic Enlargement. Moreover, we focus on the bladder smooth muscle rather than the bladder epithelium. Therefore, the results of the previous paper could not be reflected in our report.

Thank you again for your comment on our paper. I trust that the revised manuscript is suitable for publication.

Reviewer 2 Report

This study investigated the potential role of blueberries in prevention of bladder dysfunction by attenuating chronic oxidative stress. To this aim, the authors evaluated the effect of a diet containing blueberry powder in rats with bladder outlet obstruction (BOO) in the development of bladder dysfunction. Eighteen 8-week-old male rats were randomly divided in three experimental groups: a) control group which underwent to a sham surgery and given a normal diet; b) N-BOO group which underwent to a BOO surgery and given a normal diet; c) B-BOO which underwent to a BOO surgery and blueberry diet (normal diet + blueberry powder). Outcomes included oxidative stress measured by urinary markers and histological analysis of bladder rats. Compared with the N-BOO group, B-BOO rats showed a significant decreased level of one out two urinary markers investigated giving an indication of a decreased oxidative stress.

The manuscript is well structured and written.

Few aspects need to be addressed by the authors.

1. I suggest the authors to better synthesize the main results of the study (two or three sentences) at the beginning of the discussion section.

2. The authors did not describe limitations and strengths of the study. Please, add a paragraph in the discussion describing these important topics.

Author Response

Response to Reviewer 2:

We wish to express our appreciation to the Reviewer 2 for the insightful comments, which have helped us significantly improve the paper.

Comment 1: I suggest the authors to better synthesize the main results of the study (two or three sentences) at the beginning of the discussion section.

Response: We thank the Reviewer for this pertinent comment. In accordance with the Reviewer’s comment, we have added the following text in the Discussion from (p. 6, line 196-198):

“In the B-BOO group, the urinary 8-OHdG level was markedly decreased compared with that in the N-BOO group. The intake of blueberries significantly reduced the increase in bladder weight, fibrosis of the smooth muscle layer and Cx43 expression level in BOO rats.”

Comment 2: The authors did not describe limitations and strengths of the study. Please, add a paragraph in the discussion describing these important topics.

Response: In accordance with the Reviewer’s comment, we had added the following text in the discussion from (p. 7, line 249-253):

“In clinical practice, urinary urgency is important as a main complaint from patients, and it has a great influence on the QOL of patients. As this study was based on an animal model, one limitation of this study is that urinary urgency could not be evaluated. However, we believe that animal model studies can be used to investigate pathological mechanisms in detail and contribute to the discovery of new preventive and therapeutic targets.”

We wish to thank the Reviewer again for the valuable comments.